# Alteration of Substrate Specificity and Transglucosylation Activity of GH13_31 α-Glucosidase from *Bacillus* sp. AHU2216 through Site-Directed Mutagenesis of Asn258 on β→α Loop 5

**DOI:** 10.3390/molecules28073109

**Published:** 2023-03-30

**Authors:** Waraporn Auiewiriyanukul, Wataru Saburi, Tomoya Ota, Jian Yu, Koji Kato, Min Yao, Haruhide Mori

**Affiliations:** 1Research Faculty of Agriculture, Hokkaido Unifversity, Sapporo 060-8589, Japan; auiewi.waraporn@gmail.com (W.A.);; 2Faculty of Advanced Life Science, Hokkaido University, Sapporo 060-0810, Japan

**Keywords:** α-glucosidase, glycoside hydrolase family 13, transglycosylation, substrate specificity, site-directed mutagenesis, X-ray crystallography

## Abstract

α-Glucosidase catalyzes the hydrolysis of α-d-glucosides and transglucosylation. *Bacillus* sp. AHU2216 α-glucosidase (BspAG13_31A), belonging to the glycoside hydrolase family 13 subfamily 31, specifically cleaves α-(1→4)-glucosidic linkages and shows high disaccharide specificity. We showed previously that the maltose moiety of maltotriose (G3) and maltotetraose (G4), covering subsites +1 and +2 of BspAG13_31A, adopts a less stable conformation than the global minimum energy conformation. This unstable d-glucosyl conformation likely arises from steric hindrance by Asn258 on β→α loop 5 of the catalytic (β/α)_8_-barrel. In this study, Asn258 mutants of BspAG13_31A were enzymatically and structurally analyzed. N258G/P mutations significantly enhanced trisaccharide specificity. The N258P mutation also enhanced the activity toward sucrose and produced erlose from sucrose through transglucosylation. N258G showed a higher specificity to transglucosylation with *p*-nitrophenyl α-d-glucopyranoside and maltose than the wild type. E256Q/N258G and E258Q/N258P structures in complex with G3 revealed that the maltose moiety of G3 bound at subsites +1 and +2 adopted a relaxed conformation, whereas a less stable conformation was taken in E256Q. This structural difference suggests that stabilizing the G3 conformation enhances trisaccharide specificity. The E256Q/N258G-G3 complex formed an additional hydrogen bond between Met229 and the d-glucose residue of G3 in subsite +2, and this interaction may enhance transglucosylation.

## 1. Introduction

α-Glucosidase (EC 3.2.1.20; AGase) hydrolyzes α-glucosidic linkages at the non-reducing end of substrates to produce α-d-glucose [1]. This enzyme is a key enzyme in amylolytic pathways of various organisms. In addition to hydrolysis, AGase catalyzes transglucosylation by transferring a d-glucosyl residue to an acceptor. This activity is used widely to synthesize oligosaccharides and α-d-glucosides [2,3,4]. Isomaltooligosaccharides and nigerooligosaccharides, which have beneficial physiological functions [5,6,7], are industrially produced from maltooligosaccharides through transglucosylation catalyzed by fungal AGases [2,3]. The specificity of AGases toward substrate chain length and glucosidic linkages is diverse, and AGases are classified into three groups based on their substrate specificities [8]: enzymes in group I prefer heterogeneous substrates (e.g., sucrose and aryl α-d-glucosides) to homogeneous substrates such as maltose (G2), whereas enzymes in groups II and III are specific to homogeneous substrates. Enzymes in group III have a high activity toward long-chain substrates.

BspAG13_31A from a soil bacterium *Bacillus* sp. AHU2216 is an AGase that specifically acts on α-(1→4)-d-glucosidic linkages in both hydrolysis and transglucosylation reactions [9]. BspAG13_31A prefers G2 as the substrate among maltooligosaccharides, whereas most AGases show the highest activity toward maltotriose (G3) [10,11]. In addition, BspAG13_31A shows a high transglucosylation activity toward G2, even at low concentrations. In the sequence-based classification of glycoside hydrolases, BspAG13_31A belongs to glycoside hydrolase family 13 (GH13) [12]. This family is one of the largest GH families and includes a wide variety of amylolytic enzymes such as α-amylase (EC 3.2.1.1), cyclodextrin glucanotransferase (EC 2.4.1.19), branching enzyme (EC 2.4.1.18), and AGase [13,14]. This large GH13 is subdivided into 46 subfamilies based on phylogenetic analysis [15,16], and subfamily 31 (GH13_31) includes AGases and enzymes acting on α-glucosides, such as oligo-1,6-glucosidase (EC 3.2.1.10), dextran glucosidase, and isomaltulose synthase (EC 5.4.99.11). These GH13_31 enzymes include enzymes with distinct substrate specificities: α-(1→4)-linkage-specific enzymes (AGase) and α-(1→6)-linkage-specific enzymes (oligo-1,6-glucosidase and dextran glucosidase).

BspAG13_31A has three domains common to GH13 enzymes: domain A, a catalytic domain that adopts a (β/α)_8_-barrel fold; domain B, a long loop connecting the third β-strand and third α-helix of domain A (β→α loop 3); domain C, a domain formed by two antiparallel β-sheets that follows domain A [9] (Figure 1). The protruding β→α loop 8 of domain A, including two α-helices, is considered as domain B′ and forms part of the pocket-shaped active site together with domain B and β→α loop 4. These structural features are commonly found in solved structures of GH13_31 enzymes [17,18,19,20]. Our structural analysis of BspAG13_31A in complex with maltooligosaccharides (PDB entry, 5ZCC, 5ZCD and 5ZCE) revealed that His203 and Asn258 on β→α loops 4 and 5, respectively, form hydrogen bonds with the d-glucosyl residue of maltooligosaccharides in subsite +1 [9]. The maltose moiety of G3 and maltotetraose (G4), covering subsites +1 and +2 of this enzyme, adopts a less stable conformation than the global minimum energy conformation, presumably because of steric hindrance by Asn258 at subsite +2 [9]. In several GH13 AGases-type enzymes, the length of β→α loop 4 of domain A correlates with the substrate chain-length specificity (enzymes preferring short- and long-chain substrates have long and short β→α loop 4, respectively [21,22]). However, β→α loop 4 of BspAG13_31A is as long as those of trisaccharide-specific AGases, and its disaccharide specificity cannot be explained by the β→α loop 4 length. From the structure of the complex with maltooligosaccharides, we hypothesized that the binding of the d-glucosyl residue to subsite +2 in an unstable orientation is responsible for the disaccharide specificity of BspAG13_31A.

In this study, site-directed mutations were introduced at Asn258, and Asn258 mutants of BspAG13_31A were enzymatically and structurally characterized. This study revealed one of the molecular bases that diversify the substrate and reaction specificity of GH13_31 enzymes, including α-glucosidase-type enzymes with various substrate (linkage and chain-length) and reaction (hydrolysis and transglucosylation) specificities.

## 2. Results

### 2.1. Kinetic Analysis of the Reactions of BspAG13_31A Asn258 Mutants with p-Nitrophenyl α-d-Glucopyranoside and G2

BspAG13_31A Asn258 mutants, N258D, N258G, N258L, N258P, N258W, and N258Y, were prepared in the same manner as the wild-type enzyme [9], and yields of 68.9–114 mg of the mutant enzymes were achieved from 1 L of *Escherichia coli* transformant cultures. Reaction rates of the Asn258 mutants for aglycon releasing (total reaction rate) and the hydrolysis of *p*-nitrophenyl α-d-glucopyranoside (pNPG) were measured to evaluate reaction specificity. The reaction velocities followed hydrolysis-transglucosylation kinetics, and kinetic parameters were determined (Table 1). The *k*_cat1_/*K*_m1_ values, equivalent to *k*_cat_/*K*_m_ from the Michaelis–Menten equation, of the mutant enzymes were 1.6–53-fold higher than that of the wild-type enzyme. N258P showed the highest *k*_cat1_/*K*_m1_ among the *Bsp*AG13_31A variants. The *K*_TG_, giving 50% of r_tg_ (ratio of transglucosylation rate in the total reaction rate), for *p*NPG varied among the mutants. N258G and N258Y showed a 3.1- and 1.5-fold lower *K*_TG_ than the wild type did, whereas the other mutants showed 1.8–6.7-fold higher values.

The activity of Asn258 mutants toward G2 was evaluated by measuring the velocities for hydrolysis and transglucosylation, *v*_h_ and *v*_tg_, respectively, in the presence of 10 mM G2, and r_tg_ values were determined (Table 2). All mutants showed a lower *v*_h_ and *v*_tg_ than the wild type did. The *v*_h_ and *v*_tg_ of the mutant enzymes were 0.19–7.1% and 0.0017–20.6% of those of the wild type, respectively. The r_tg_ value of N258G (89.6%) was 3-fold higher than that of the wild type (30.1%), whereas the r_tg_ for N258P (37.8%) was similar to that of the wild type. The r_tg_ of the other mutants was low (3.61–4.60%), indicating that these mutants predominantly catalyzed hydrolysis differently to wild-type BspAG13_31A.

Reaction velocities of BspAG13_31A Asn258 mutants using various concentrations of G2 were measured. Reaction rates of N258G and N258P obeyed the rate equation based on the kinetic model in which both hydrolysis and transglucosylation occur, and those of the other mutants obeyed the Michaelis–Menten equation. The kinetic parameters of the mutants were determined (Table 3). The *k*_cat_/*K*_m_ values of the mutant enzymes (*k*_cat1_/*K*_m1_ of E258G and E258P) were 23–1590-fold lower than *k*_cat1_/*K*_m1_ values of the wild-type enzyme. These values were 1.8–91-fold lower than *k*_cat1_/*K*_m1_ for pNPG, revealing that the mutant enzymes, especially N258P/W/Y, showed a low preference for G2 over pNPG. The *K*_TG_ value of N258G for G2 was 19.1-fold lower than that of the wild type, whereas that of N258P was similar to that of the wild type. N258G showed lower *K*_TG_ values for both G2 and *p*NPG than the wild type did, indicating that the N258G mutation enhanced the specificity to transglucosylation.

### 2.2. Reactions of BspAG13_31A Asn258 Mutants with Various Disaccharides

Reaction rates of Asn258 mutants to 1 mM G2, trehalose, kojibiose, nigerose, isomaltose, and sucrose were measured (Figure 2). All mutant enzymes displayed weak activity toward glucobioses other than G2 and showed high specificity to the α-(1→4)-d-glucosidic linkage, as observed for the wild type. In addition, velocities of N258L, N258P, and N258W to sucrose relative to maltose increased considerably: the reaction rates of these mutants to sucrose were 33%, 88%, and 24% of those to G2, respectively. These relative reaction rates were much higher than that of the wild-type enzyme (0.19%). In particular, the reaction rate of N258P (3.39 s^–1^) was 19-fold higher than that of the wild type (0.179 s^–1^). N258L, and N258P, and wild-type reaction rates were measured at various sucrose concentrations (0.1–25 mM). These rates obeyed the Michaelis–Menten equation under the analytical conditions used, and kinetic parameters for sucrose were determined (Table 4). The *k*_cat_/*K*_m_ values of these enzymes were 29–87% of *k*_cat1_/*K*_m1_ for G2 (*k*_cat_/*K*_m_ in N258L), whereas that of the wild type was only 0.0993%. 

The transglucosylation of sucrose by N258L and N258P was examined (Figure 3). In the reaction using the wild type and N258L with 100 mM sucrose as the substrate, only hydrolytic products, d-fructose and d-glucose, were detected. However, N258P synthesized a transglucosylation product in the early stage of the reaction. Analysis of the reaction products using high-performance anion exchange chromatography with pulsed amperometric detection (HPAEC-PAD) indicated that the reaction mixture contained a product that showed the same retention time as authentic erlose (α-maltosyl β-fructoside), together with d-glucose and d-fructose. This observation indicates that N258P catalyzed the α-(1→4)-d-glucosyl transfer to sucrose.

### 2.3. Substrate Chain-Length Specificity of BspAG13_31A Asn258 Mutants

The substrate chain-length specificity of the Asn258 mutants was evaluated using a series of maltooligosaccharides (Figure 4). Reaction rates of all mutants to G3 and longer maltooligosaccharides obeyed the Michaelis–Menten equation. N258D/L/Y preferred G2 to G3, as observed for the wild-type enzyme, whereas N258G/P had a significantly higher *k*_cat_/*K*_m_ to G3 than to G2. The *k*_cat_/*K*_m_ values of N258G and N258P for G3 were 11- and 27-fold higher than the *k*_cat1_/*K*_m1_ values for G2, respectively, whereas *k*_cat_/*K*_m_ of the wild type for G3 was only 26% of the *k*_cat1_/*K*_m1_ for G2. The *k*_cat_/*K*_m_ values of N258G and N258P for G3 were 1.8- and 3.5-fold higher than that of the wild type, respectively, whereas the other mutants showed much lower *k*_cat_/*K*_m_ values for all maltooligosaccharides tested compared with those of the wild type. The *k*_cat_/*K*_m_ values of N258G and N258P for maltooligosaccharides longer than those of G3 decreased with an increasing number of d-glucosyl residues. 

### 2.4. Structural Analysis of BspAG13_31A E256Q/N258G and E256Q/N258P

We solved the crystal structures of inactive double mutants, E256Q/N258G and E256Q/N258P, in which the general acid/base catalyst Glu256 was substituted with Gln together with the mutation at Asn258, in complex with G3. Crystals of E256Q/N258G and E256Q/N258P were prepared by co-crystallization in the presence of 5 mM G3 and G4, respectively, and the structures of E256Q/N258G and E256Q/N258P were solved at 1.7 Å and 1.5 Å resolutions, respectively (Table 5). The structural comparison showed that the structures of the double mutants were very similar to the structure of the parent enzyme E256Q in complex with G3 (PDB entry, 5ZCD). The root-mean-square deviation of Cα atoms between E256Q/N258G and E256Q was 0.138 Å, and that between E256Q/N258P and E256Q was 0.143 Å. As in the structural analysis of the E256Q mutant in complex with G3, in which the structure of β→α loop 6 of domain A (Phe282–Pro295) was not determined, because of poor electron density [9], the electron density of the corresponding regions in the double mutants was also poor, and part of this loop was not modeled (Lys290–Glu293 in N256Q/N258G and Leu287–Lys294 in N256Q/N258P).

The *F*_o_–*F*_c_ maps of E256Q/N258G and E256Q/N258P clearly revealed the existence of a substrate molecule in the substrate binding sites (Figure 5). In the analysis of the co-crystal of E256Q/N258P and G4, the electron density of only three d-glucosyl residues covering subsites from –1 to +2 was observed. The G3-bound structure of E256Q/N258P was not solved by co-crystallization with G3 (data not shown). Orientations of the d-glucosyl residues of G3 in subsites –1 and +1 were similar in E256Q/N258G, E256Q/N258P, and E256Q (PDB entry, 5ZCD) (Figure 5C,D). The conformation of the d-glucose residue in subsite +2 of the double-mutant enzymes was similar but differed from that found in E256Q. Torsion angles of d-glucosyl residues in subsites +1 and +2 of E256Q/N258G and E256Q/N258P were close to the torsion angles of d-glucosyl residues in the stable α-glucan: Φ (5-O, 1-C, 4′-O, 4′C) = 102° and Ψ (1-C, 4′O, 4′-C, 5′-C) = –120° [23] (Table 6), which is considered the global minimum energy structure [24]. In E256Q/N258G, the SD of Met229 located at the C-terminus of β→α loop 4 is suitably located to form a hydrogen bond with 3-O of the d-glucose residue in subsite +2. This hydrogen bonding interaction was not observed in the complexes of E256Q and E256Q/N258P. The orientation of the side chain of Phe282 in E256Q/N258G, E256Q/N258P, and E256Q differed, presumably because of the closely located amino acid residue at position 258 (Gly, Pro, and Asn in E256Q/N258G, E256Q/N258P, and E256Q, respectively). The side chain of Phe282 of E256Q/N258G and E256Q was suitably positioned to interact with the d-glucosyl residue at subsite +2, whereas that of Phe282 in E256Q/N258P was distal from the corresponding d-glucosyl residue and unlikely to have any contact with the substrate. In addition, the carbonyl oxygen of Phe282 was suitably positioned to form a water-mediated hydrogen bond with the d-glucose residue at subsite +2 in E256Q/N258G and E256Q but not in E256Q/N258P.

## 3. Discussion

AGase displays various substrate specificities in terms of d-glucosidic linkage and substrate chain length [1]. This enzyme also catalyzes transglucosylation, which is useful for the industrial production of oligosaccharides and α-d-glucosides. Thus, understanding the molecular basis for recognizing substrates and regulating transglucosylation activity provides valuable information that should facilitate AGase applications. In this study, the substrate chain-length specificity and transglucosylation activity of GH13_31 AGase and BspAG13_31A were modulated through site-directed mutagenesis at Asn258, and structures of the mutant enzymes in complex with the substrate G3 were determined via X-ray crystallography. Asn258 is located at β→α loop 5 of domain A and suitable to form a hydrogen bond with 2-O of the d-glucosyl residue in subsite +1. Eliminating this hydrogen bond by substituting Asn258 with Leu, which has a similar side chain size, caused a large reduction in *k*_cat_/*K*_m_ to G2 but not to pNPG (Table 1 and Table 2). This result indicates that the hydrogen bond provided by Asn258 is important for the reaction with G2. The ΔΔG of the N258L mutation for the reaction with G2, calculated from Equation (1) [30], was 15 kJ/mol, supporting the role that Asn258 forms a hydrogen bond with the substrate.
ΔΔG = −RT[ln(*k*_cat_/*K*_m_)_N258L_ − ln(*k*_cat1_/*K*_m1_)_wild-type_](1)

As the N258D mutation significantly reduced the *k*_cat_/*K*_m_ to G2, a negative charge at the Asn258 position is unsuitable for substrate binding by BspAG13_31A. The introduction of bulkier amino acids, Tyr and Trp, also caused an apparent reduction in *k*_cat_/*K*_m_ by these mutants to G2. This loss in activity is presumably because these bulky residues sterically hindered G2 binding. This steric hindrance also likely hindered acceptor binding, reducing the transglucosylation activity of N258W and N258Y toward pNPG and G2. In contrast to the reaction with G2, N258W and N258Y showed a 3.2–3.3-fold higher *k*_cat1_/*K*_m1_ to pNPG than that of the wild type. Introducing aromatic residues may afford favorable interactions with the pNP group at subsite +1 and enhance the activity toward pNPG. N258G and N258P showed a lower activity to G2 than the wild type did, presumably because the hydrogen bond interaction with the substrate in subsite +1 is absent, as discussed above. Nonetheless, these mutant enzymes showed a 1.8- and 3.5-fold higher *k*_cat_/*K*_m_ to G3 than the wild type did (Figure 4). The affinities in subsite +2 (*A*_+2_) of N258G and N258P, estimated using Equation (2) [31], were 6.1 and 8.5 kJ/mol, respectively, whereas the *A*_+2_ of the wild type was −3.5 kJ/mol.
*A*_+2_ = RT[ln(*k*_cat_/*K*_m_)_G3_ − ln(*k*_cat1_/*K*_m1_)_G2_](2)

Structural analysis of E256Q/N258G and E256Q/N258P in complex with G3 revealed that the conformation of the maltose moiety bound at subsites +1 and +2 of the double-mutant enzymes was different from that of E256Q (Figure 5). Torsion angles of the d-glucosyl residues in subsites +1 and +2 of these mutants were close to those of d-glucosyl residues in a stable α-glucan [23]. A molecular mechanics study of maltose suggested that torsion angles of d-glucosyl residues at subsite +1 and +2 in the E256Q-G3 complex are less stable than the global minimum [24], and the conformation of G3 observed in E256Q/N258G and E256Q/N258P is predicted to be more thermodynamically stable than that in E256Q. This conformational difference of G3 presumably arises from the elimination of steric hindrance by Asn258. In the E256Q/N258P-G3 complex, interactions between Phe282 and the d-glucosyl residue in subsite +2 are absent, whereas these interactions were observed in E256Q/N258G and E256Q (Figure 5). Moreover, any additional interactions through the N258P mutation were not observed at subsite +2. Therefore, the stabilized conformation of G3 likely enhanced trisaccharide specificity rather than increasing the enzyme–substrate interaction. Torsion angles of d-glucosyl residues at subsites +1 and +2 of other GH13 enzymes, acting on α-(1→4)-glucan, are also similar to those of stable α-glucan (Table 6), suggesting that these enzymes recognize substrates in a stable conformation. In the E256Q/N258G-G3 complex, an additional hydrogen bond between Met229 and 3-O of the d-glucose residue at subsite +2 is predicted (Figure 5). This hydrogen bond may stabilize the Michaelis complex and stabilize the transition state because the *K*_m_ of N258G was 25-fold lower for G3 when compared with that of the wild type. This interaction may also be favorable for acceptor binding in transglucosylation, and N258G showed a higher preference for transglucosylation than the wild type did (Table 1, Table 2 and Table 3). N258P displayed a 29-fold higher *k*_cat_/*K*_m_ for sucrose than the wild type did (Table 4). As N258L also showed a high preference for sucrose, a hydrophobic interaction may favor the reaction with sucrose. In contrast to the wild type and N258L, which only catalyzed hydrolysis in the reaction with 100 mM sucrose, N258P produced erlose by transglucosylation. These results indicated that Pro at the 258th position facilitates the binding of d-fructosyl residues of an acceptor sucrose at subsite +2. An α-(1→4)-d-glucosidic linkage was specifically formed even in the reaction with sucrose. This Pro mutation may be useful for creating a biocatalyst to produce oligosaccharides from sucrose.

## 4. Materials and Methods

### 4.1. Preparation of BspAG13_31A Mutants

The expression plasmids for BspAG13_31A mutants were constructed using the Primestar Mutagenesis Basal Kit (Takara Bio; Shiga, Japan). The primers for Asn258 single mutants were: 5′-GAGGCAX_1_X_2_X_3_GGTGTAACATCGGACAGG-3′ (sense; X_1_X_2_X_3_: GGT, CCG, TGG, CTG, TAT, and GAT for N258G, N258P, N258W, N258L, N258Y, and N258D, respectively) and 5′-TACACCX_4_X_5_X_6_TGCCTCTCCAACCGTCAT-3′ (antisense; X_4_X_5_X_6_: ACC, CGG, CCA, CAG, ATA, and ATC for N258G, N258P, N258W, N258L, N258Y, and N258D, respectively). For the inactive double mutants, E256Q/N258G and E256Q/N258P, 5′-GTTGGACAGGCAX_7_X_8_X_9_GGTGTAACATCGGAC-3′ (sense; X_7_X_8_X_9_: GGT and CCG for E256Q/N258G and E256Q/N258P, respectively) and 5′-GCTGTCCGATGTTACACCX_10_X_11_X_12_TGCCTGTCC-3′ (antisense; X_10_X_11_X_12_: ACC and CGG for E256Q/N258G and E256Q/N258P, respectively) were used. Expression plasmids of wild-type BspAG13_31A and the E256Q mutant derived from the pET-23a vector (Novagen, Darmstadt, Germany) [9] were used as templates for the single and double mutants, respectively. Mutant enzymes were prepared in the same manner as the wild type [9]. *E. coli* BL21(DE3) transformants carrying the expression plasmid were grown in 1 L of LB medium containing 100 µg·mL^–1^ of ampicillin at 37 °C until *A*_600_ reached 0.5. Recombinant protein production was induced by adding isopropyl β-thio-d-galactopyranoside to a final concentration of 0.1 mM, and growth was continued at 18 °C for 20 h. The recombinant enzyme was purified from the cell-free extract using a Ni-immobilized Chelating Sepharose Fast Flow column (2.0 cm i.d. × 5.0 cm, GE Healthcare, Uppsala, Sweden). Pooled fractions were dialyzed against 10 mM sodium phosphate buffer (pH 7.0). For crystallization, the double mutants were further purified by DEAE Sepharose Fast Flow column chromatography (2.7 cm i.d. × 36 cm; GE Healthcare) in 10 mM sodium phosphate buffer (pH 7.0). Adsorbed protein was eluted by a linear 0–0.5 M NaCl gradient. Collected fractions were dialyzed against 10 mM Tris-HCl buffer (pH 7.0). The protein concentration was determined by amino acid analysis using the theoretical amino acid composition of the protein.

### 4.2. Kinetic Analysis of Reactions with Various Substrates

The substrate specificity of BspAG13_31A Asn258 mutants was examined by measuring reaction velocities using the following substrates at 1 mM: G2 (Nihon Shokuhin Kako, Tokyo, Japan) and G4; G3 (Nacalai Tesque, Kyoto, Japan); maltopentaose (G5, Wako Pure Chemical Industries, Osaka, Japan), maltohexaose (G6), maltoheptaose (G7), and kojibiose; trehalose (Hayashibara, Okayama, Japan) and nigerose; isomaltose (Tokyo Chemical Industry, Tokyo, Japan); sucrose (Kanto Chemical, Tokyo, Japan). A reaction mixture (100 μL) containing an appropriate enzyme concentration, 1 mM substrate, 42 mM sodium phosphate buffer (pH 7.0), and 0.02 mg^–1^·mL^–1^ of bovine serum albumin (BSA) was incubated at 37 °C for 10 min. Enzyme reactions were terminated by adding 50 μL of 4 M TrisHCl buffer (pH 7.0), and the d-glucose concentration was measured by the reported photometric method [9].

In the reaction with pNPG and G2 (in N258G and N258P), in which hydrolysis and transglucosylation simultaneously occur, kinetic parameters for pNPG and G2 were determined from velocities for aglycone release (*v*_ag_) and d-glucose release (*v*_glc_; *v*_glc_ = 2*v*_h_ + *v*_tg_), respectively, at various substrate concentrations ([S]). The *v*_ag_ and *v*_glc_ are expressed by Equations (3) and (4), respectively [32,33]:*v*_ag_ = (*k*_cat2_[S]^2^ + *k*_cat1_*K*_m2_[S])/([S]^2^ + *K*_m2_[S] + *K*_m1_*K*_m2_),(3)
*v*_glc_ = (*k*_cat2_[S]^2^ + 2*k*_cat1_*K*_m2_[S])/([S]^2^ + *K*_m2_[S] + *K*_m1_*K*_m2_),(4)

Kinetic parameters *k*_cat1_, *k*_cat2_, *K*_m1_, and *K*_m2_ are defined as follows (Equations (5)–(8)) by the rate constants shown in Figure 6:*k*_cat1_ = *k*_1_*k*_2_*k*_3_(*k*_−4_ + *k*_5_)/{*k*_4_*k*_5_(*k*_−1_ + *k*_2_) + *k*_1_(*k*_−4_ + *k*_5_)(*k*_2_ + *k*_3_)},(5)
*k*_cat2_ = *k*_2_*k*_5_/(*k*_2_ + *k*_5_),(6)
*K*_m1_ = *k*_3_(*k*_−1_ + *k*_2_)(*k*_−4_ + *k*_5_)/{*k*_4_*k*_5_(*k*_−1_ + *k*_2_) + *k*_1_(*k*_−4_ + *k*_5_)(*k*_2_ + *k*_3_)},(7)
*K*_m2_ = {*k*_4_*k*_5_(*k*_−1_ + *k*_2_) + *k*_1_(*k*_−4_ + *k*_5_)(*k*_2_ + *k*_3_)}/{*k*_1_*k*_4_(*k*_2_ + *k*_5_)},(8)

The transglucosylation ratio (r_tg_) follows Equation (9).
r_tg_ = *v*_tg_/*v*_ag_,(9)

As *v*_tg_ is expressed by Equation (10), r_tg_ is also expressed by Equation (11).
*v*_ag_ = *k*_cat2_[S]^2^/([S]^2^ + *K*_m2_[S] + *K*_m1_*K*_m2_),(10)
r_tg_ = [S]/(*K*_TG_ + [S]),(11)
where *K*_TG_ is the transglycosylation parameter (substrate concentration that gives r_tg_ 50%) and is expressed by Equation (12).
*K*_TG_ = *k*_cat1_*K*_m2_/*k*_cat2_,(12)

A reaction mixture (50 μL) containing an appropriate concentration of enzymes, 0.2–100 mM substrate, 42 mM sodium phosphate buffer (pH 7.0), and 0.02 mg^–1^·mL^–1^ of BSA was incubated at 37 °C for 10 min. The reaction was stopped by adding 100 μL of 2 M Tris-HCl (pH 7.0) or 1 M Na_2_CO_3_ for the reaction with maltose or pNPG, respectively. d-Glucose and *p*-nitrophenol liberated from G2 and pNPG, respectively, were measured, as described previously [9]. Non-linear regression of the rate equations to reaction rates was carried out using Grafit version 7.0.2 (Erithacus Software, East Grinstead, UK).

Kinetic parameters for G2 (Asn258 variants other than N258G and N258P), G3–G7, and sucrose were determined by fitting the Michaelis–Menten equation to the hydrolytic velocity (*v*_h_) at substrate concentrations of 0.1–100 mM using Grafit version 7.0.2. Enzyme reactions were performed as the reaction with G2.

The r_tg_ for the reaction with 10 mM G2 was determined from *v*_glc_ and *v*_tg_ (velocity for G3 formation) using Equation (9). These reaction rates were measured by HPAEC-PAD (Dionex ICS-5000+, Thermo Fisher Scientific, Waltham, MA, USA). A reaction mixture (1 mL) containing an appropriate concentration of enzyme, 10 mM G2, 42 mM sodium phosphate buffer (pH 7.0), and 0.02 mg^–1^·mL^–1^ of BSA was incubated at 37 °C. Aliquots of the reaction mixture (0.2 mL) were taken every 3 min, and the reaction was stopped by heating at 100 °C for 5 min. The reaction products were quantified by HPAEC-PAD as follows: injection volume, 10 μL; column, Carbopac PA1 (4 mm i.d. × 250 mm; Thermo Fischer Scientific); eluent, 640 mM NaOH; flow rate, 0.8 mL/min. d-Glucose and G3 (10–100 µM) were used as standards.

### 4.3. Analysis of the Reaction Product from Sucrose

A reaction mixture (1 mL) containing enzymes (9.8 μM wild type, 0.8 μM N258P, or 12 μM N258L), 100 mM sucrose, and 10 mM MES-NaOH buffer (pH 6.0) was incubated at 37 °C. Aliquots (0.14 mL) were taken from the reaction mixture at indicated times and heated at 100 °C for 5 min. Five micrograms of carbohydrate was spotted on TLC plate silica gel 60 F_254_ (Merck, Darmstadt, Germany). A developing solvent, chloroform/acetic acid/water (6/7/1, *v*/*v*/*v*), was used and the carbohydrates were visualized by spraying with the detection reagent, 85% orthophosphoric acid/0.01% aniline in acetone (10/1, *v*/*v*), and heating the TLC plate.

The transglucosylation product from sucrose by N258P was analyzed by HPAEC-PAD. A reaction mixture (0.5 mL) containing 0.49 μM N258P, 100 mM sucrose, and 10 mM MES-NaOH buffer (pH 6.0) was incubated at 37 °C for 30 min. The reaction was stopped by heating at 100 °C for 5 min. HPAEC-PAD analysis was performed as described above, but 480 mM NaOH was used as the eluant. d-Glucose, d-fructose (Nacalai Tesque), sucrose, and erlose (Seikagaku, Tokyo, Japan) (200 μM) were used as standards.

### 4.4. Crystal Structure Analysis of BspAG13_31A Mutants

Crystals of BspAG13_31A E256Q/N258G and E256Q/N258P in complex with maltooligosaccharides were obtained by the hanging drop vapor diffusion method, in which 1 μL of protein solution (29 mg·mL^–1^) was mixed with the same volume of a mixture containing 10 mM G3 or G4, 0.2 M CaCl_2_, and 20% polyethylene glycol 3350, and the samples were incubated for 1 week at 20 °C. The crystals were directly selected from the crystallization solution and flash-cooled for X-ray diffraction experiments. Diffraction data of the BspAG13_31A mutants were collected on beamline BL41XU at SPring-8 (Hyogo, Japan). The datasets were indexed, integrated, scaled, and merged using the XDS program suite [34]. All data collection statistics are summarized in Table 5.

The structures of BspAG13_31A mutants were determined by the molecular replacement method with the program Auto BL-41XU MR in the Phenix program package [35,36]. The structure of BspAG13_31A E256Q in complex with G3 (PDB entry, 5ZCD) was used as the search model. Several rounds of refinement were carried out using the program Phenix.refine in the Phenix program suite, alternating between manual fitting and rebuilding based on 2*F*_o_−*F*_c_ and *F*_o_−*F*_c_ electron densities using COOT [35,37]. Water molecules, Ca^2+^, and substrates were built using 2*F*_o_−*F*_c_ and *F*_o_−*F*_c_ electron densities. The final refinement statistics and geometry defined by MolProbity [38] are shown in Table 5. The atomic coordinates and structure factors have been deposited in the Protein Data Bank (http://wwpdb.org/; PDB entries 8IBK and 8IDS). Figures of structures were prepared by PyMol ver. 2.5.0 (Schrödinger, LLC, New York, NY, USA).

## 5. Conclusions

In this study, Asn258 of BspAG13_31A was shown to be a key residue for disaccharide specificity. Stabilizing the conformation of G3 by eliminating the steric hindrance caused by Asn258 enhanced the trisaccharide specificity and preference for transglucosylation. The disaccharide specificity of BspAG13_31A is explained by the binding of G3 and longer maltoligosaccharides to the substrate binding site in an unstable conformation. BspAG13_31A adapts a mechanism for disaccharide specificity different from that in *Hallomonas* sp. GH13_23 disaccharide-specific AGase, which has a long β→α loop 4 to prohibit the binding of trisaccharides and longer oligosaccharides to the substrate binding site.

## Figures and Tables

**Figure 1 molecules-28-03109-f001:**
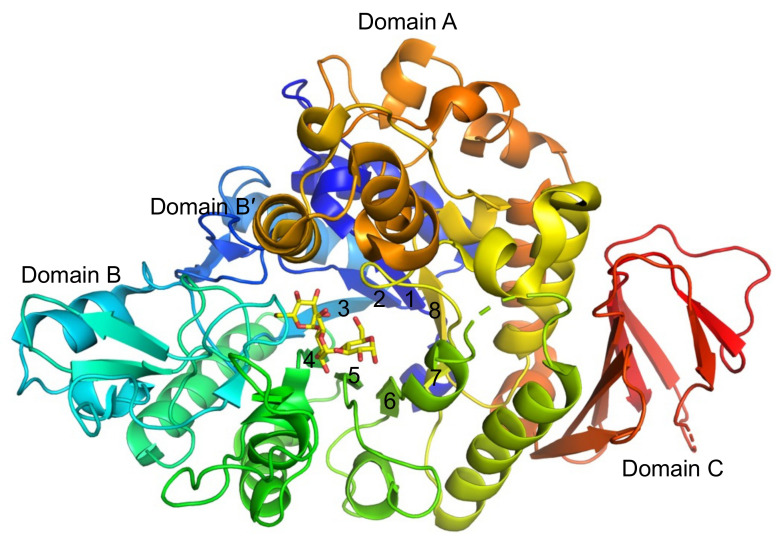
Overall structure of BspAG13_31A. The structure of the BspAG13_31A E256Q mutant in complex with G3 (PDB entry, 5ZCD) is shown in rainbow coloring from the N-terminus (blue) to C-terminus (red). Yellow sticks represent G3 bound to subsites −1 to +2. Numbers indicate the numbering of the β-strands, forming domain A.

**Figure 2 molecules-28-03109-f002:**
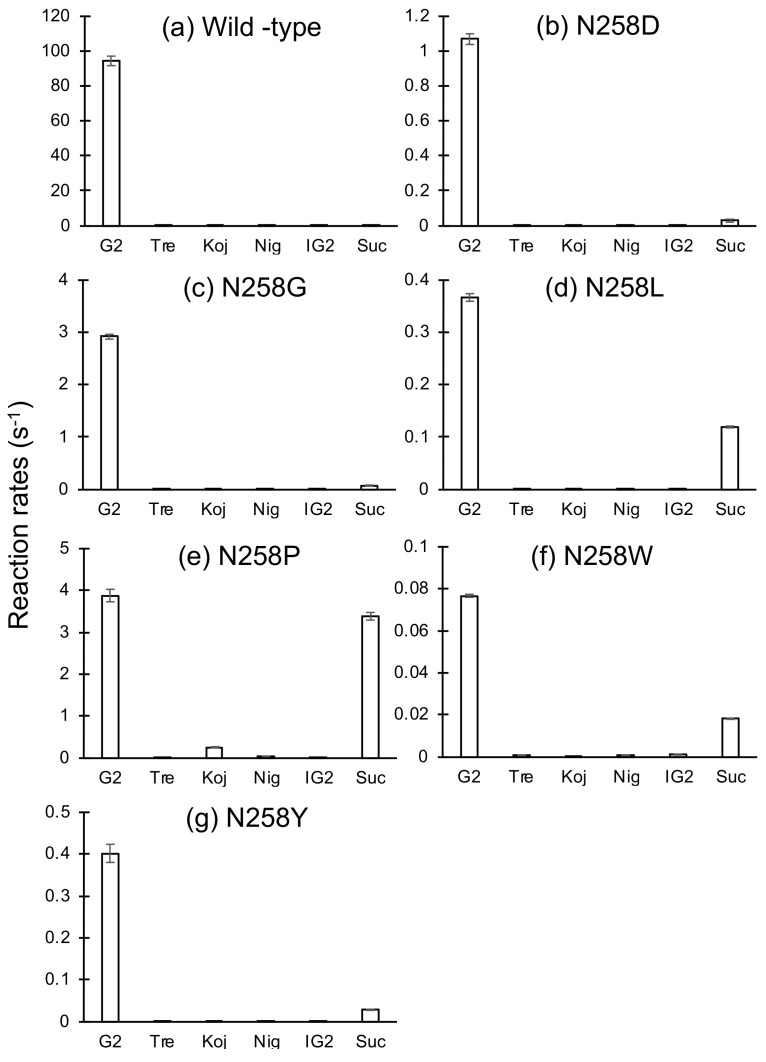
Reaction rates of Asn258 mutants of BspAG13_31A. (**a**) Wild type, (**b**) N258D, (**c**) N258G, (**d**) N258L, (**e**) N258P, (**f**) N258W, and (**g**) N258Y. Reaction rates to 1 mM substrates are shown. Values are the average ± standard deviation from three independent experiments. Panel (**a**) was made using data from reference [9].

**Figure 3 molecules-28-03109-f003:**
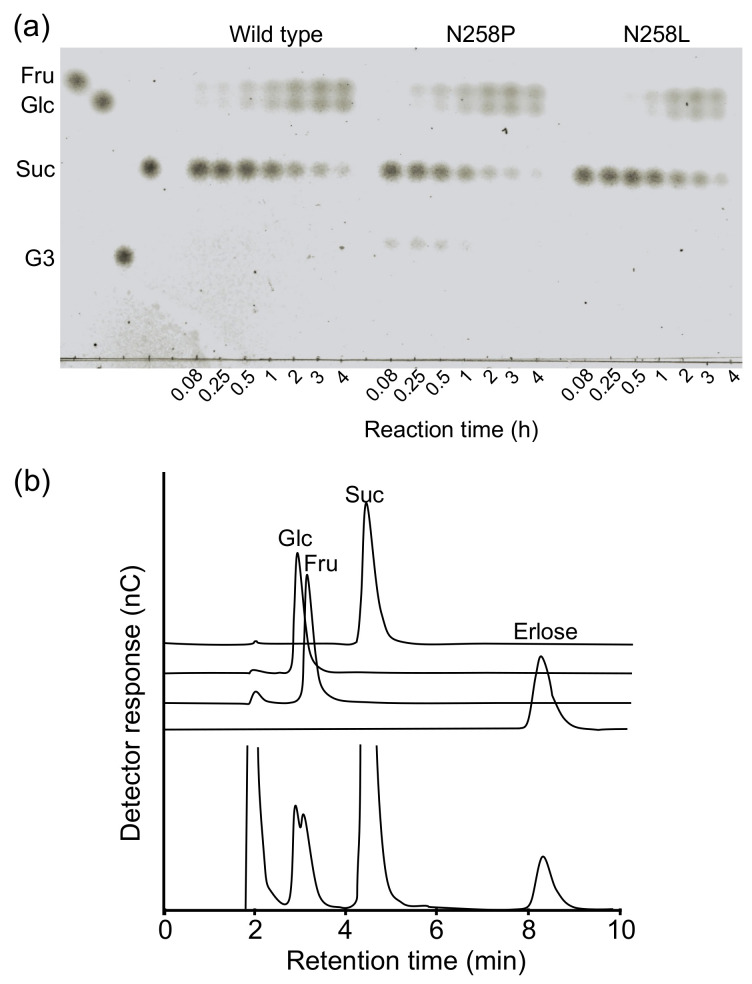
Analysis of the reaction products from sucrose. (**a**) TLC analysis of the reaction products from sucrose by wild type, N258P, and N258L. The reaction time is shown below the figure. (**b**) HPAEC-PAD analysis of the reaction products produced by N258P after a 30 min reaction.

**Figure 4 molecules-28-03109-f004:**
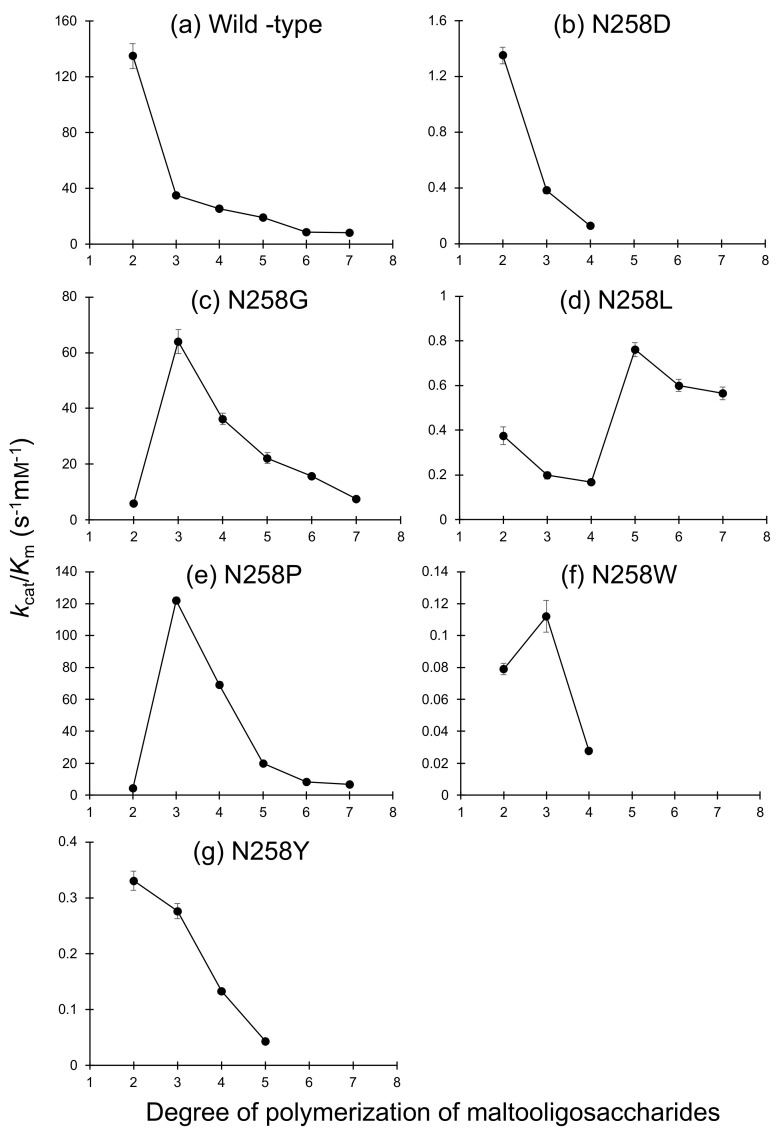
Substrate chain-length specificity of Asn258 mutants of BspAG13_31A. (**a**) Wild type, (**b**) N258D, (**c**) N258G, (**d**) N258L, (**e**) N258P, (**f**) N258W, and (**g**) N258Y. The *k*_cat_/*K*_m_ values for a series of maltooligosaccharides are shown. The values presented are the average and standard deviation from three independent experiments. The *k*_cat1_/*K*_m1_ values of wild type, N258G, and N158P for G2 are plotted. Panel (**a**) was made using data from reference [9]. The *k*_cat_/*K*_m_ values of N258D and N258W for G5–G7 and those of N258Y for G6 and G7 were not determined.

**Figure 5 molecules-28-03109-f005:**
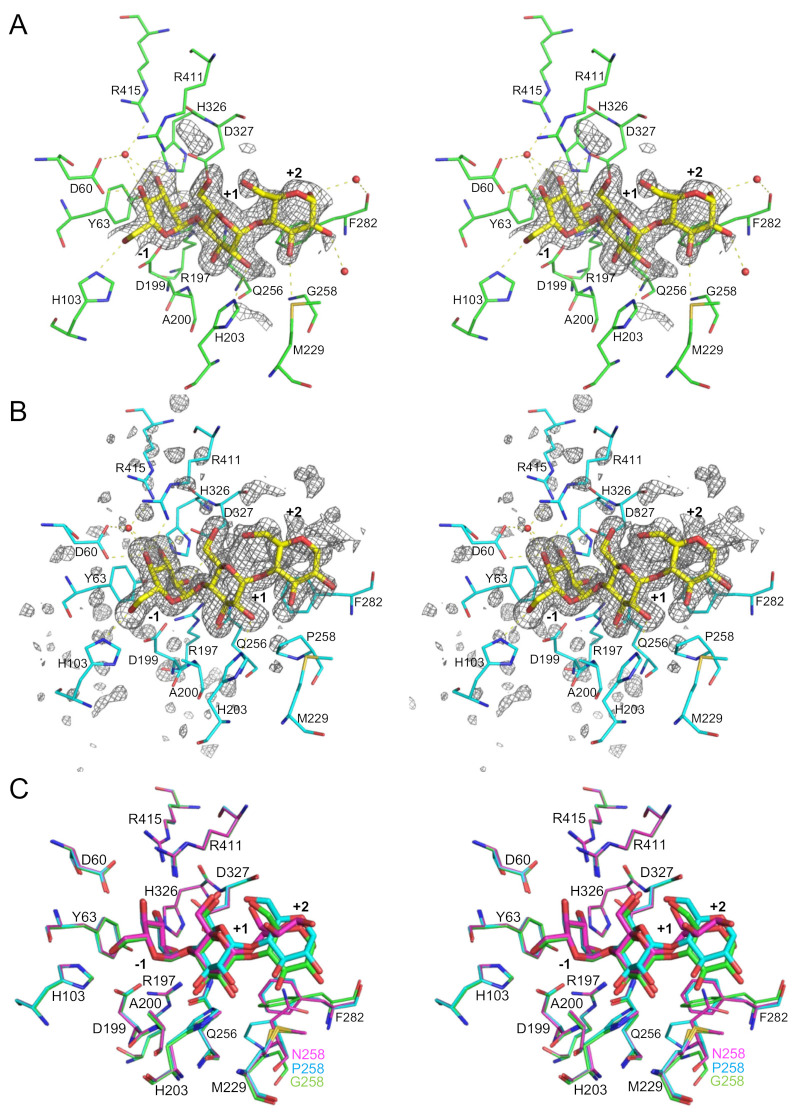
Structure of the active sites of E256Q/N258G and E256Q/N258P in complex with G3. The *F*_o_-*F*_c_ OMIT maps of E256Q/N258G (**A**) and E256Q/N258P (**B**) (contoured at 2.5 σ) are shown in stereo view. Red balls represent water molecules bound to the ligands. Ligands are shown as yellow sticks. Putative hydrogen bonds are indicated by yellow dotted lines. The subsite numbers are shown. The superimposition of E256Q, E256Q/N258G, and E256Q/N258P is shown in panel (**C**). E256Q (PDB entry, 5ZCD), E256Q/N258G, and E256Q/N258P are shown in magenta, green, and cyan stick representations, respectively. Surface models of the BspAG13_31A variants are shown in panel (**D**).

**Figure 6 molecules-28-03109-f006:**
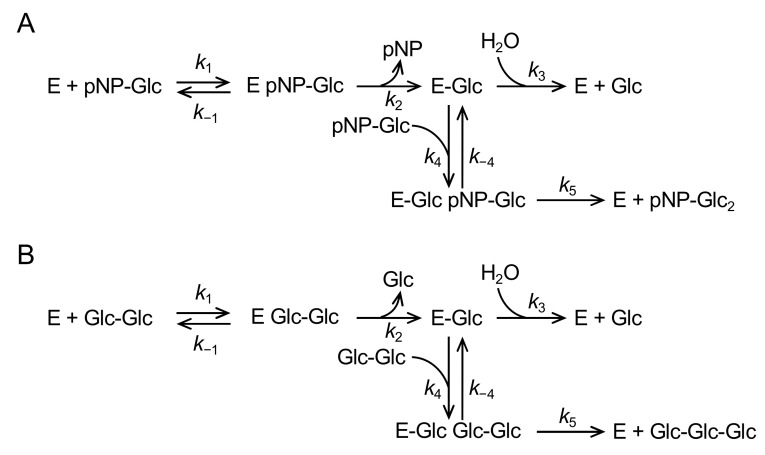
Reaction scheme for the reaction with pNPG (**A**) and G2 (**B**).

**Table 1 molecules-28-03109-t001:** Kinetic parameters of BspAG13_31A Asn258 variants to pNPG.

Enzyme	*k*_cat1_ (s^−1^)	*k*_cat2_ (s^−1^)	*K*_m1_ (mm)	*K*_m2_ (mm)	*k*_cat1_/*K*_m1_ (s^−1^mm^−1^)	*K*_TG_ (mm)
Wild type ^1^	4.72	13.9	2.05	4.71	2.30	2.48
N258D	9.88 ± 0.23	16.3 ± 1.7	1.92 ± 0.03	26.9 ± 3.5	5.15	16.6
N258G	1.95 ± 0.28	2.29 ± 0.09	0.194 ± 0.060	0.971 ± 0.211	10.0	0.792
N258L	5.61 ± 0.30	9.70 ± 0.70	1.51 ± 0.15	7.66 ± 0.71	3.72	4.43
N258P	81.3 ± 7.0	125 ± 18	0.674 ± 0.108	10.5 ± 2.8	121	6.81
N258W	1.48 ± 0.05	2.39 ± 0.22	0.193 ± 0.019	12.2 ± 1.4	7.67	7.58
N258Y	2.50 ± 0.33	8.27 ± 0.66	0.293 ± 0.006	5.17 ± 1.28	8.53	1.64

^1^ Data are quoted from reference [9]. Values are the mean ± standard deviation from three independent experiments.

**Table 2 molecules-28-03109-t002:** Reaction rates of Asn258 variants of BspAG13_31A to 10 mM G2.

Enzyme	*v*_h_ (s^−1^)	*v*_tg_ (s^−1^)	r_tg_ (%)
Wild type ^1^	193	82.9	30.1
N258D	6.72 ± 0.387	0.284 ± 0.010	4.05
N258G	2.03 ± 0.76	17.4 ± 2.3	89.6
N258L	2.02 ± 0.40	0.0975 ± 0.0289	4.60
N258P	11.3 ± 2.4	6.86 ± 1.26	37.8
N258W	0.367 ± 0.028	0.0137 ± 0.0030	3.61
N258Y	0.967 ± 0.050	0.0389 ± 0.0120	3.87

*v*_h_ and *v*_tg_ are reaction rates for hydrolysis and transglucosylation, respectively. ^1^ Data are quoted from reference [9]. Values are the mean ± standard deviation from three independent experiments.

**Table 3 molecules-28-03109-t003:** Kinetic parameters of BspAG13_31A Asn258 variants to G2.

Enzyme	*k*_cat1_ (s^−1^)	*k*_cat2_ (s^−1^)	*K*_m1_ (mm)	*K*_m2_ (mm)	*k*_cat1_/*K*_m1_ (s^−1^mm^−1^)	*K*_TG_ (mm)
Wild type ^1^	205	1890	1.52	201	135	21.6
N258D ^2^	13.6 ± 0.2	N.A.	10.2 ± 0.6	N.A.	1.33	N.A.
N258G	3.20 ± 0.20	190 ± 7	0.534 ± 0.089	66.9 ± 3.3	5.99	1.13
N258L ^2^	4.27 ± 0.09	N.A.	11.5 ± 1.4	N.A.	0.371	N.A.
N258P	46.8 ± 3.9	379 ± 86	10.5 ± 1.5	219 ± 89	4.46	27.0
N258W ^2^	0.918 ± 0.087	N.A.	10.8 ± 0.6	N.A.	0.0850	N.A.
N258Y ^2^	2.06 ± 0.01	N.A.	6.25 ± 0.62	N.A.	0.330	N.A.

^1^ Data are quoted from reference [9]. ^2^ The *k*_cat1_, *K*_m1_, and *k*_cat1_/*K*_m1_ of D258D/L/W/Y are *k*_cat_, *K*_m_, and *k*_cat_/*K*_m_ from the Michaelis–Menten equation, respectively. Values are the average ± standard deviation from three independent experiments. N.A., not applicable.

**Table 4 molecules-28-03109-t004:** Kinetic parameters of BspAG13_31A variants to sucrose.

Enzyme	*k*_cat_ (s^−1^)	*K*_m_ (mm)	*k*_cat_/*K*_m_ (s^−1^mm^−1^)
Wild type	4.58 ± 0.1	34.2 ± 2.6	0.134
N258L	1.94 ± 0.07	18.1 ± 0.5	0.107
N258P	39.2 ± 0.6	10.1 ± 0.4	3.90

Values are the average ± standard deviation from three independent experiments.

**Table 5 molecules-28-03109-t005:** Summary of data collection and refinement statistics.

	E256Q/N258G-G3	E256Q/N258P-G3
PDB ID	8IBK	8IDS
Data collection		
Beamline	Spring-8 BL41XU	Spring-8 BL41XU
Space group	*P*2_1_2_1_2_1_	*P*2_1_2_1_2_1_
Unit cell parameters a, b, c, (Å)	57.0, 90.0, 128.4	56.5, 89.2, 128.3
Wavelength (Å)	1.00	1.00
Resolution range (Å)	48.0–1.70 (1.76–1.70)	50.0–1.50 (1.59–1.50)
Total No. of reflections	964,831 (152,139)	1,374,196 (220,850)
No. of unique reflections	72,935 (11,510)	104,555 (16,645)
R_meas_ (%) *	11.2 (98.1)	8.8 (83.5)
<I/σ(I)>	17.2 (1.99)	19.7 (2.76)
CC_1/2_	0.999 (0.864)	0.999 (0.835)
Completeness (%)	99.6 (98.4)	99.9 (99.6)
Redundancy	13.2 (13.2)	13.1 (13.3)
Refinement		
R_work_/R_free_ (%) **	18.2/20.9	16.5/18.6
No. of atoms		
Macromolecules	4501	4472
Ligand/ion	36	36
Water	564	545
B-factors (Å^2^)		
Macromolecules	23.8	19.6
Ligand/ion	25.9	27.8
Water	31.5	28.8
RMSD from ideal		
Bond lengths (Å)	0.006	0.016
Bond angles (°)	0.86	1.37
Ramachandran		
Favored (%) Allowed (%) Outliers (%)	96.85 3.15 0	97.21 2.79 0

Values in parentheses are for the highest-resolution shell. * *R*_meas_ = Σ*_hkl_*{*N*(*hkl*)/[*N*(*hkl*) − 1]}^1/2^ Σ*_i_*|*I_i_*(*hkl*) − <*I*(*hkl*)>|/Σ*_hkl_*Σ*_i_ I_i_*(*hkl*), where <*I*(*hkl*)> and *N*(*hkl*) are the mean intensity of a set of equivalent reflections and multiplicity, respectively. ** *R_work_ =* Σ*_hkl_*||*F_obs_|* − *|F_calc_||/*Σ*_hkl_*|*F_obs_|*, *R*_free_ was calculated for 5% randomly selected test sets not used in the refinement.

**Table 6 molecules-28-03109-t006:** Comparison of the torsion angles of G3 bound to BspAG13_31A variants with other GH13 enzymes.

			Residues in Subsites −1 and +1		Residues in Subsites +1 and +2	
Enzyme	Origin	Ligand	Φ	Ψ	Φ	Ψ
*Bsp*AG13_31A	*Bacillus* sp. AHU2216					
E256Q ^1^		G3	42.8°	−151°	64.6°	−168°
E256QN258G		G3	31.6°	−153°	105°	−113°
E256QN258P		G3	43.9°	−156°	106°	−109°
α-Amylase E208Q ^2^	*Bacillus subtilis*	G5	28.6°	−148°	109°	−125°
α-Amylase D300N ^3^	*Homo sapiens* (pancreatic)	G6	34.6°	−148°	112°	−117°
CGTase wild type ^4^	*Bacillus circulans*	G4	25.2°	−154°	110°	−125°
Neopullulanase E357Q ^5^	*Geobacillus stearothermophilus*	G4	33.8°	−144°	92°	−112°
Amylosucrase E328Q ^6^	*Neisseria polysaccharea*	G7	33.8°	−155°	67.1°	−148°

Φ (5-O, 1-C, 4′-O, 4′C), Ψ (1-C, 4′O, 4′-C, 5′-C). CGTase, cyclodextrin glucanotransferase. References 1–6 are [9,25,26,27,28,29], respectively.

## Data Availability

The datasets generated and/or analyzed in this study are available from the corresponding author upon reasonable request.

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
