# Peer review of "Alteration of Substrate Specificity and Transglucosylation Activity of GH13_31 α-Glucosidase from Bacillus sp. AHU2216 through Site-Directed Mutagenesis of Asn258 on β→α Loop 5"

_molecules, 2023, doi:10.3390/molecules28073109_

Round 1

Reviewer 1 Report

The manuscript describes the effects of mutating a residue (Asn258) of the GH13_31 alpha-glucosidase from Bacillus. It is very interesting to see the effect different residues at this single position has on the hydrolysis as well as transglycosylation activity. The activity assays are supported by crystal structures of two of the studied mutants. The study contributes with new knowledge relevant to altering transglycosylation activity of GH13 enzymes in general, and also to understand the chain length preferences of GH13_31 enzymes.

In my opinion, the manuscript is well written. However, it would help the reader, if the authors included a (overall) structure figure, which i.e. could show the position of the beta-alpha loop 5 – this will make the rationale behind the study described in lines 60-81 easier to follow.

I would also like to see a structure figure of the two solved structures, where they are shown with a surface representation – to get an idea of active site architecture. Figure 4 is in my opinion too small. It is hard to see e.g. the hydrogen bonds.

In Figure 3 data for some of the longer maltooligosaccharides are missing for some of the mutants. Is this due to too low activity, or was the activity on those not analysed? Maybe include a note stating the reason.

Consider to refer to your previously published “wt” structure using the PDB code in relevant place(s). Right now, you only refer to it in the materials and methods as a search model.

Author Response

Thank you for your fruitful comments and suggestions. We sincerely revised our manuscript. Responses to your comments are as follows:

In my opinion, the manuscript is well written. However, it would help the reader, if the authors included a (overall) structure figure, which i.e. could show the position of the beta-alpha loop 5 – this will make the rationale behind the study described in lines 60-81 easier to follow.

-->According to your comment, we added Figure 1 showing overall structure of BspAG13_31A. Numbering of β-strand is also indicated.

I would also like to see a structure figure of the two solved structures, where they are shown with a surface representation – to get an idea of active site architecture. Figure 4 is in my opinion too small. It is hard to see e.g. the hydrogen bonds.

-->Figure 5 in the revised manuscript is shown larger than the previous one. Surface representation figure is added as panel D.

In Figure 3 data for some of the longer maltooligosaccharides are missing for some of the mutants. Is this due to too low activity, or was the activity on those not analysed? Maybe include a note stating the reason.

-->Data missing in Figure 4 (3 in the previous version) were not analyzed. We noted this in the revised manuscript.

Consider to refer to your previously published “wt” structure using the PDB code in relevant place(s). Right now, you only refer to it in the materials and methods as a search model.

-->We added citations of wt (E256Q in complex with G3) structure using PDB code in relevant places.

Reviewer 2 Report

Line 67: Please include PDB no. for structural analysis.

Table 1: Authors should explain briefly about kcat1, kcat2, Km1, Km2, KTG

Did authors perform the wild-type experiment along with other mutants? This will be more accurate.

Figure 1, 3: It will be easier to read, if the name of the wild-type and mutant enzymes are added in these Figures.

Line 210: Typo? “…..and that between E256Q/N258P and E256Q?”

Figure 4: add PDB no. of E256Q

Author Response

Thank you for your fruitful comments and suggestions. We are sorry that the previous manuscript included typo as you pointed out. Responses to your comments are as follows:

Line 67: Please include PDB no. for structural analysis.

--> We added PDB codes to this sentence according to your comment.

Table 1: Authors should explain briefly about kcat1, kcat2, Km1, Km2, KTG

--> The kinetic parameters are explained in the Materials and Methods section.

Did authors perform the wild-type experiment along with other mutants? This will be more accurate.

-->Kinetic data of the wild type only for the reaction with sucrose were obtained in this study. Other data are quoted from reference 9. This is mentioned in figure captions.

Figure 1, 3: It will be easier to read, if the name of the wild-type and mutant enzymes are added in these Figures.

--> Name of the wild type and mutant enzymes are added to panels of Figure 2 and 4 in the revised manuscript.

Line 210: Typo? “…..and that between E256Q/N258P and E256Q?”

-->I’m sorry it was typo. It is corrected.

Figure 4: add PDB no. of E256Q

-->The PDB number is added to the figure caption of Figure 5 of the revised manuscript.